# *ClNAC100* Is a NAC Transcription Factor of Chinese Fir in Response to Phosphate Starvation

**DOI:** 10.3390/ijms241310486

**Published:** 2023-06-22

**Authors:** Yuxuan Zhao, Shuotian Huang, Lihui Wei, Meng Li, Tingting Cai, Xiangqing Ma, Peng Shuai

**Affiliations:** 1College of Forestry, Fujian Agriculture and Forestry University, Fuzhou 350002, China; 1200428022@fafu.edu.cn (Y.Z.); 1210421003@fafu.edu.cn (S.H.); 5220422082@fafu.edu.cn (L.W.); m18059086636@163.com (M.L.); c15705981830@163.com (T.C.); lxymxq@126.com (X.M.); 2Chinese Fir Engineering Technology Research Center of the State Forestry and Grassland Administration, Fuzhou 350002, China

**Keywords:** *Cunninghamia lanceolata*, phosphate starvation, NAC, hormone, stress

## Abstract

Phosphate (Pi) deficiency is one of the most limiting factors for Chinese fir growth and production. Moreover, continuous cultivation of Chinese fir for multiple generations led to the reduction of soil nutrients, which hindered the yield of Chinese fir in southern China. Although NAC (NAM, ATAF, and CUC) transcription factors (TFs) play critical roles in plant development and abiotic stress resistance, it is still unclear how they regulate the response of Chinese fir to phosphate (Pi) starvation. Based on Pi-deficient transcriptome data of Chinses fir root, we identified a NAC transcription factor with increased expression under Pi deficiency, which was obtained by PCR and named *ClNAC100*. RT-qPCR confirmed that the expression of *ClNAC100* in the root of Chinese fir was induced by phosphate deficiency and showed a dynamic change with time. It was positively regulated by ABA and negatively regulated by JA, and *ClNAC100* was highly expressed in the roots and leaves of Chinese fir. Transcriptional activation assay confirmed that *ClNAC100* was a transcriptional activator. The promoter of *ClNAC100* was obtained by genome walking, which was predicted to contain a large number of stress, hormone, and growth-related cis-elements. Tobacco infection was used to verify the activity of the promoter, and the core promoter was located between −1519 bp and −589 bp. We identified 18 proteins bound to the *ClNAC100* promoter and 5 ClNAC100 interacting proteins by yeast one-hybrid and yeast two-hybrid, respectively. We speculated that AHL and TIFY family transcription factors, calmodulin, and E3 ubiquitin ligase in these proteins might be important phosphorus-related proteins. These results provide a basis for the further study of the regulatory mechanism and pathways of *ClNAC100* under Pi starvation.

## 1. Introduction

Chinese fir (*Cunninghamia lanceolata* (Lamb.) Hook), as an important fast-growing and high-yield tree species, is widely cultivated in southern China. However, the concentration of available phosphate (Pi) in the soil of the Chinese fir cultivation area is low, and the continuous plantation of Chinese fir leads to the decrease of soil nutrients and environmental pollution caused by excessive application of phosphate fertilizer [1,2]. These factors affect the growth and yield of Chinese fir directly. Plants have evolved complex physiological and molecular mechanisms to cope with Pi starvation, optimizing the high-affinity acquisition, translocation, utilization, and control of Pi homeostasis [3,4,5]. It has been reported that Chinese fir adapted to Pi starvation by adjusting root morphological traits to improve Pi acquisition and effective utilization [6]. The study found that the Pi mobilization capacity of Chinese fir decreased with stand age, but the Pi recycling capacity within Chinese fir increased [7]. Although the response of Chinese fir to phosphorus starvation has been studied in many ways, the underlying molecular mechanism by which Chinese fir defends against Pi deficiency still needs to be elucidated.

Under the condition of Pi starvation, plants adapt their physiology, biochemistry, and morphology to Pi starvation by regulating the expression of various genes [8]. Among them, transcription factors (TFs) play an important role in activating transcription or inhibiting genes. Studies have shown that the MYB family, WRKY family, bHLH family, NAC family, and other transcription factors directly or indirectly participate in the transcriptional regulation of Pi starvation [9,10,11]. Phosphate starvation response (PHR) proteins, as MYB-family members, play a central role in transcriptional regulatory responses in response to Pi starvation [12]. PHR- and SPX-domain-containing proteins form a Pi signaling network to regulate plant Pi homeostasis [5,11]. Other MYB family members are also involved in transcriptional regulation in response to Pi starvation, such as *MYB2*, *MYB4*, and *MYB62* [13,14,15]. The WRKY family is also a transcription factor frequently appearing in plant adaptation studies to Pi starvation, such as a result of the study suggesting that *WRKY75* is a modulator of Pi starvation responses as well as root development [16]. Meanwhile, the WRKY45 transcription factor activates PHT1;1 expression in response to phosphate starvation which enhanced Pi intake in Arabidopsis [17]. Therefore, one of the important ways to improve the yield of Chinese fir is to explore the transcriptional factors in response to low-phosphate stress and identify their functions to explore their regulatory pathways in response to low-phosphate stress.

A family of transcription factors comprises NAC (NAM, ATAF, CUC1/2) protein domains widely distributed in plants; they are also an important class of factors that regulate plant response to stress [18,19]. These transcription factors generally have a highly conserved domain at the N-terminus of the protein and a variable activation domain at the C-terminus [20,21]. NAC transcription factors play important roles in plant growth and development. For example, Xie found that the *NAC1* gene negatively regulates *E2Fa* gene expression to control root growth in Arabidopsis [22]. In addition, the regulatory function of NAC in plant response to drought, salt stress, and low temperature has been widely studied. *CsNAC1* is strongly induced by drought stress in the leaves of *Citrus reshni* and *Citrus limonia*, and also by salt stress, cold, and ABA in the leaves and roots of *Citrus reshni* [23]. Among the NAC transcription factors associated with cold stress, several are involved in the regulation of plant cold tolerance through CBF-COR signaling [24].

It is worth noting that a study explains how *OsNAC016* is involved in the regulatory pathway of Pi starvation: as the downstream target gene of *OsPHR1* and *OsPHR4*, *OsNAC016* can activate the expression of the downstream gene *OsSPX2*, which plays a negative regulatory role in rice adaptation to Pi starvation [25]. Another study found that *ANAC044* was induced by phosphorus deficiency, and worked as a transcriptional repressor that inhibits the induction of ethylene biosynthesis by down-regulating the expression of *ACS6* and *ACO1* under Pi starvation, thus inhibiting the ethylene-mediated cell wall Pi reutilization in Arabidopsis root [26]. Before that, though NAC family TFs have been mentioned in the studies of plant responses to low-phosphate stress and appear in many differentially expressed genes of the transcriptome on phosphate deficiency, the regulatory mechanisms by which NAC transcription factors modulate Pi absorption or translocation are still unclear, especially in gymnosperms.

Here, we reported that a significantly differentially expressed NAC transcription factor named *ClNAC100*, which was cloned from the phosphate-deficient transcripts of Chinese fir root, and its promoter were obtained. We conducted the bioinformatics analysis of this transcription factor and its promoter and preliminarily screened out the protein-binding *ClNAC100* promoter and the protein interacting with this transcription factor through the experiment. These results can better help us to study the regulatory pathway of *ClNAC100* response to Pi starvation.

## 2. Results

### 2.1. Cloning and Bioinformatics Analysis of ClNAC100 Gene from Chinese Fir

Based on the transcriptome data of Chinese fir with phosphate deficiency, specific primers were designed, and the PCR products were sequenced to obtain the NAC gene sequence Appendix A. By contrast, it was found that the NAC gene had the closest relationship with *AtNAC100* in Arabidopsis which is from the NAM family, so it was named *ClNAC100*. It has the typical NAC-family-conserved domain NAM, and this gene was identified as an NAC family gene (Figure 1a). The protein encoded by the *ClNAC100* gene had a total of 415 amino acids, and its molecular weight was 46,861.65. The protein’s isoelectric point was 6.72, the instability coefficient was 51.13, and the average hydrophilicity (GRAVY) was −0.723, indicating that the protein was classified as unstable. ClNAC100 protein has no signal peptide and transmembrane structure (Figure 1b,c).

ClNAC100 was compared with amino acid sequences of homologous genes of eight species of plants, including *Arabidopsis thaliana*, *Tamarix hispida*, *Liquidambar formosana*, *Zea mays*, *Gossypium arboreum*, *Larix kaempferi*, *Taxus chinensis*, and *Camellia lanceoleosa*, to explore the similarity of ClNAC100 sequence with other species. The results are shown in Figure 1d. It was found that these sequences had a highly conserved NAC domain in the N-terminal region, and a diverse array of different amino acid sequences in the C-terminal region, which was consistent with the structure of NAC protein. In addition, the C terminus also contained some sequences with high homology, indicating that its function also had a certain similarity.

### 2.2. ClNAC100 Transactivation Activity Analysis

To verify whether the ClNAC100 protein had transcriptional activation, the full-length sequence of ClNAC100, the NAC domain (N-terminal sequence) from 1 to 169 amino acids (aa), and the TR region (C-terminal sequence) from 170 to 415 were inserted into the pGBKT7 vector (Figure 2a). When the yeast containing four different plasmids was transferred to SD/-Trp/-His/-Ade triple-deficient medium containing X-α-gal, the control empty plasmid pGBKT7 and the recombinant plasmid inserted with N-terminal fragment could not grow normally and showed no chromogenic reaction. The recombinant plasmid containing the full-length and C-terminal fragments of the inserted gene grew normally and showed a blue color (Figure 2b). The results showed that the protein encoded by the *ClNAC100* gene had transcription activation activity, and the transcription activation domain was located at the C-terminus, which belonged to the transcription activator with transcription activation activity.

### 2.3. ClNAC100 Expression Pattern Analysis

To verify the accuracy of transcriptome data, we performed RT-qPCR to verify the expression of *ClNAC100*, and the results showed that its expression was increased; it was consistent with the expression trend in the transcriptome, and the expression level was 3.17 times that of CK (Figure 3a).

In the RT-qPCR experiment, the expression of ClNAC100 in the roots of Chinese fir seedlings was significantly increased at 2 h after Pi deficiency treatment, which was 10.3 times that of the normal Pi supply (CK). The expression of *ClNAC100* decreased to 2.7 times that of CK after 4 h of treatment. The expression of *ClNAC100* increased during the 12 h–3 d interval of P deficiency treatment, and the expression reached the highest level at 3 d. The expression of *ClNAC100* decreased with the time of phosphorus deprivation treatment, but it was still higher than CK. Pi deficiency induced the up-regulation of *ClNAC100* expression in the root of fir and showed dynamic changes with time (Figure 3b).

The transcriptional expression of *ClNAC100* may be induced by some hormones. Exogenous hormones ABA and JA were applied to the roots of Chinese fir. The expression of *ClNAC100* was analyzed by RT-qPCR. After treatment for 3 h, it reached 1.62 times that of CK. After 6 h of treatment, the expression of *ClNAC100* was significantly higher than that of CK, and its expression was 2.02 times that of CK. After 12 h of treatment, the expression of *ClNAC100* was down. When the roots were treated with JA for 1 h, the expression of *ClNAC100* was significantly down, only 0.49 times that of CK. The expression of *ClNAC100* decreased continuously at 3 h and 6 h, and reached the lowest value at 6 h, and the expression was only 0.41 times that of CK. These results indicated that the application of ABA could induce *ClNAC100* expression (Figure 3c), while the application of JA could inhibit *ClNAC100* expression (Figure 3d) in the roots of Chinese fir treated with exogenous hormones.

The expression level of *ClNAC100* was different in different parts. Because the *ClNAC100* was found in Chinese fir roots, its relative expression level in roots was used as a control. The part-specific expression of *ClNAC100* was analyzed in the root, stem, and leaf of Chinese fir seedlings (Figure 3e). The results showed that the expression level of *ClNAC100* in the root and leaf was significantly higher than that in the stem. In conclusion, *ClNAC100* was specifically expressed in different parts of Chinese fir.

### 2.4. Promoter Cloning and Activity Analysis of ClNAC100

By using the PCR-based genome walking procedure, we obtained about 2200 bp promoter fragment upstream of the start codon of the *ClNAC100* gene Appendix A. We predicted two TSSS (Table 1), located 1882 bp and 800 bp upstream of the translation start codon, respectively. In addition, by prediction, it was found that there were a large number of hormone response elements (Appendix A) such as ABA response elements: AAGAA-motif and ABRE on the *ClNAC100* promoter. Meja response elements: CGTCA-motif and TGACG-motif as well as a salicylic acid response element as-1, ethylene response element ERE, and auxin-related element GC-motif. In addition, there were some light-responsive elements (I-box, GT1-motif), stress- and damage-related elements (STRE, LTR, MYB, WUN-motif), and stress-related transcription factor binding sites (W-box, MYC, and MBS) on the promoter. Furthermore, we performed a visual analysis of these cis-elements (Figure 4a).

The four promoter fragments were cloned and replaced the 35S promoter of the PBI121 vector. The PBI121 empty vector was used as a positive control, and tobacco leaves without treatment were used as a negative control (Figure 4b). The results of GUS staining showed that five different deletion promoter fragments could drive the expression of the GUS gene (Figure 4c), and the leaves of P1-PBI121 infection turned light blue. The leaves of infected P2-PBI121 and P3-PBI121 were mostly dark blue, and P3-PBI121 was darker. Only a small number of extremely light blue or blue dots could be observed on tobacco leaves infected with P4-PBI121. Combined with the above TSS and cis-element prediction results, the *ClNAC100* promoter has the strongest activity in the −1053 bp to −589 bp part and the *ClNAC100* TSS is located in this part, which may contain key elements to enhance the promoter activity and drive the high expression of GUS gene in the −1519 bp to −589 bp region. Additionally, the promoter of other fragments was significantly different. However, the promoter from −589 bp to −1 bp showed only weak activity, indicating that there was no element to drive gene expression.

### 2.5. Combined with ClNAC100 Gene Promoter Protein Screening

Based on the predicted distribution of cis-elements on the *ClNAC100* gene promoter, the promoter was divided into three small fragments Pro1, Pro2, and Pro3 (Figure 5a), which were inserted upstream of the AbAir reporter gene in the pAbAi vector. The recombinant vector was linearized and integrated into the YIHGold yeast genome (Figure 5b). The minimum concentrations (Figure 6a) of AbA inhibition for Pro1, Pro2, and Pro3 bait strains were determined to be 600 ng/mL, 800 ng/mL, and 200 ng/mL, respectively. To reduce the occurrence of false positives, the following experiments were performed with AbA at concentrations of 700 ng/mL, 900 ng/mL, and 300 ng/mL to screen the library of Pro1, Pro2, and Pro3 bait strains.

The monoclonal colonies’ PCR products were detected by 1% gel electrophoresis (Figure 6b). Finally, 18 proteins were identified (Table 2), including 2 TFs. *ClNAC100* was derived from the AHL family related to plant protein phosphorylation and stress (Protein-No.14) and the TIFY family related to the jasmonate-induced signaling pathway (Protein-No.16), respectively, indicating that *ClNAC100* was the downstream target gene of the above two transcription factors. In addition, a calcium-binding protein (protein-No.6) was found to bind to the *ClNAC100* promoter.

### 2.6. Screening of ClNAC100 Interacting Proteins

By the Mating method, the Y2H yeast strain transfected with the bait plasmid was used to interact with a qualified Chinese fir yeast cDNA library, which was preliminarily screened by SD/-Trp/-Leu plate and rescreened by SD/-Trp/-Leu/-His/-Ade/X-α-gal plate. The blue monoclonal colonies with good growth were selected for PCR, and the PCR products were detected by 1% gel electrophoresis. After screening and comparison, we finally obtained five proteins (Table 3) annotated as heavy metal-related isoprenylated plant protein, protein translation factor, E3 ubiquitin-protein ligase, and potassium transporter.

## 3. Discussion

*ClNAC100* was found to have the highest similarity with *AtNAC100* of the NAM family in Arabidopsis. The functional study of *AtNAC100* in Arabidopsis showed that overexpression of *AtNAC100* could improve the tolerance of Arabidopsis to low iron, which increased the primary root length, lateral root number, and iron reductase activity under low iron treatment [27]. There may be some functional similarities between *ClNAC100* and *AtNAC100*.

*ClNAC100* was up-regulated in the roots of Chinese fir under Pi deficiency, and other NAC transcription factors found to be related to Pi deficiency stress, such as *OsNAC016* and *ANAC044*, were induced to be up-regulated in roots under this stress, although they were involved in different regulatory mechanisms [25,26]. The ClNAC100 changed dynamically with time in the roots of Chinese fir treated with Pi deficiency. Plant physiology, biochemistry, metabolic activities, and morphogenesis vary at different times under Pi deficiency [28,29,30]; we speculate that the differential expression of the *ClNAC100* gene at different time points may be involved in different regulatory functions.

In plants, the promoter of NAC family genes contains a large number of hormone-responsive elements and their expression can be induced by hormones. RT-qPCR results showed that ABA increased the expression of *ClNAC100* in the roots of Chinese fir, while JA decreased its expression. In other plants, for example, the expression of NAC transcription factors in Arabidopsis is induced by different hormones, and these hormones regulate the expression of NAC transcription factors in a time-specific manner [31]. ABA treatment specifically induced *StNAC053* expression in potato seedlings [32]. Exogenous MeJA induced a decrease in the expression of most NAM family members in rice [33]. A large number of studies have shown that hormones are involved in the regulation network of Pi starvation of plants, especially in the uptake and utilization of Pi by root growth and development [34,35]. It was possible that *ClNAC100* played an important role in the response of Chinese fir to Pi starvation and the hormone–Pi regulation pathway.

Cis-element prediction results show that the *ClNAC100* promoter contains many hormone response elements, stress-related response elements, and transcription factor binding sites, which is consistent with previous studies on stress-related NAC gene promoters [31,36,37]. The basic condition of the eukaryotic promoter driving gene expression is that there is a core promoter element, which contains sequence modules such as TATA-box, initiator (Inr), TFⅡB recognition element (BRE), and core promoter downstream element (DPE). GUS staining results showed that the promoter activity between −1519 bp and −589 bp was the strongest. Combined with the TSS prediction results, we suggest that this region contains *ClNAC100* core promoter elements and a large number of other important elements.

Despite the presence of a large number of recognition sequences for transcription factors in the NAC promoter, it is not clear which transcription factors can drive the transcription of NAC TF. Two transcription factors that recognize the *ClNAC100* promoter were identified by yeast one-hybrid analysis, one of which belongs to the AHL family, a class of transcription factors containing the AT-hook motif. Previous studies have shown that the *AHL* gene is involved in plant growth, development, and response to stress [38]. It was found that most *AHL* genes were expressed in soybean roots and mainly mediated stress responses in roots [39]. Wong also found that *HAI1* is a protein phosphatase involved in abiotic stress and ABA signaling, which can directly dephosphorylate *AHL10* protein to participate in the regulation of stress or ABA-induced pathways [40]. Chinese fir may regulate ClNAC100 transcription by AHL for growth or response to stress, but the specific function remains to be verified.

The other belongs to the TIFY family. Besides regulating the development of plants, this family of transcription factors is also widely involved in plant stress responses, such as drought, salt stress, osmotic stress, and signal transduction network regulation of various hormones, especially JA [41,42,43]. The TIFY family members are the transcription factors regulating the MeJA-induced flavonoid biosynthesis, meanwhile, PcMYB10 and PcMYC2 can directly interact with each other and with JAZ proteins, which participate in the transcriptional regulation of jasmonate-mediated flavonoid biosynthesis [44]. In addition, the accumulation of flavonoid compounds is closely related to Pi deficiency in plants [45]. The root system is an important organ for plants to sense Pi starvation and plant hormone signal transduction allows plants to better adapt to stress. *ClNAC100* in the roots of Chinses fir was negatively regulated by JA; we speculated that this transcription factor might regulate ClNAC100 in the joint regulation pathway of JA-Pi starvation.

In addition, the *ClNAC100* promoter binds to calmodulin, and some calmodulins in Arabidopsis can bind to DNA [46]. The *PRAD* gene of *Arabidopsis thaliana* can participate in the rhizosphere acidification induced by low phosphate, and CAM5 can interact with PRAD to enhance its transcriptional activation. The mutation of CAM5 leads to the weakening of the rhizosphere acidification induced by low phosphate, indicating that CAM5 plays an important role in the regulation of this pathway [47]. Zhu found that, in the presence of the calmodulin-binding transcription factor *CAMTA*, calmodulin CML24 could induce the expression of *ALMT1* and promote malate secretion in roots to respond to stress [48]. We hypothesized that this protein was a DNA-binding calmodulin or could co-regulate the expression of *ClNAC100* with other proteins.

Protein ubiquitination and degradation pathways are ubiquitous and play an important role in plant life activities. Protein ubiquitination is generally completed by ubiquitin-activating enzyme E1, ubiquitin-conjugating enzyme E2, and ubiquitin-ligase enzyme E3. Some studies have shown that protein ubiquitination is widely involved in molecular regulation in response to Pi starvation. For example, *ATL8*, a RING E3 ligase, modulates phosphate homeostasis in Arabidopsis; the relative expression levels of the genes involved in the maintenance of Pi homeostasis are differentially modulated in mutant and overexpressed plants compared with the WT under different Pi regimes [49]. Arabidopsis NLA protein belongs to E3 ubiquitinase and interacts with BUC24 to maintain plant Pi homeostasis [50]. E3 ligase PRU1 can interact with WRKY6, and PUR1 can ubiquitinate and degrade WRKY6 under phosphate starvation conditions, relieve the inhibition of WRKY6 on the expression of PHO1, and improve the ability of Arabidopsis to transport Pi under low phosphate stress [51]. NAC protein is also often regulated by ubiquitination. The E3 ligase BRUTUS facilitates degradation of VOZ1/2 transcription factors, especially under drought and cold stress conditions [52]. An E3 ubiquitin ligase was identified in the yeast two-hybrid assay, indicating that ClNAC100 is subject to ubiquitination and degradation, inhibiting its regulation of downstream target genes. Whether ClNAC100 is ubiquitylated in regulatory pathways in response to Pi starvation requires further verification.

## 4. Materials and Methods

### 4.1. Plant Materials and Treatments

The plant materials were 1-year-old Chinese fir seedlings of “061 clones” from Yangkou Forest farm in Fujian Province. The roots of light substrate seedlings were washed and planted in the washed river sand and treated with Hoagland nutrient solution with 1/3 concentration and 1 mM KH_2_PO_4_ phosphate for 10 days.

Wild-type tobacco (*Nicotiana benthamiana*) was cultured in an incubator until 3–4 weeks old with a temperature of 24 °C, the humidity of 60–70%, light intensity of 5000 lx, 16 h light/8 h dark light.

### 4.2. Cloning of the ClNAC100

The roots of Chinese fir with good growth after seedling survival were selected for liquid nitrogen quick-freezing, and total RNA was extracted according to the method provided by TIANGEN RNAprep Pure Plant Plus Kit (Beijing, China). The first Strand cDNA was synthesized using NovoScript^®^ Plus All-in-one 1st Strand cDNA Synthesis SuperMix (Suzhou, China). According to the transcriptome data of phosphate deficiency in Chinese fir root (Accession number: PRJNA964476), the specific primers ClNAC100-F/ClNAC100-R (Appendix A) were designed, and the cDNA sequence with complete ORF was obtained with PCR. After gel electrophoresis verification and gel purification, the pMD-18 vector was connected to transform DH5α, and positive clones were selected for propagation and sent to the sequencing company for sequencing.

### 4.3. Bioinformatics Analysis

The proteins of *ClNAC100* were aligned by TAIR, and the protein domain was analyzed by SMART. ProtParam was used for the *ClNAC100* gene encoding protein molecular weight, isoelectric point, unstable factor, and physical and chemical properties such as hydrophilicity. Online analysis software SignalP4.1 was used for the analysis and prediction of signal peptides. TMHMM 2.0 was used to measure the transmembrane domain of the protein. The amino acid sequence of the ClNAC100 homologous gene of Chinese fir was downloaded from NCBI public database. The amino acid sequences of Chinese fir ClNAC100 and the above plants were arranged in the format, and Clustal X2.1 software was used for multi-sequence comparison.

### 4.4. Transactivation Activity Analysis

Two fragments of N-NAC (NAC domain), C-NAC (TR-region), and intact ClNAC100 protein divided according to the ClNAC100 protein domain were fused with the GAL4 DNA binding domain. According to the instructions for ClonExpress^®^II One Step Cloning Kit (Nanjing, China), a specific primer (Appendix A) was designed to insert the full-length gene or fragment between the 5 ‘-EcoRI and BamHI-3′ cleavage sites of vector pGBKT7. The recombinant vector and pGBKT7 transformed into Y2HGold yeast receptive state were coated on SD/-Trp plate and cultured inversely at 30 °C for 3 days. The 2–3 mm well-grown monoclonal yeasts were applied to the SD/-Trp/-His/-Ade/X-α-gal plate. Yeast growth and color were observed.

### 4.5. ClNAC100 Expression Pattern Analysis

The Chinese fir plants with good growth after seedling survival were selected for phosphate deficiency treatment for 7 days. The solution was configured as 1/3 concentration of Hoagland nutrient solution without phosphate, and 1 L was taken and added into 1 mM KH_2_PO_4_ solution for normal phosphate supply treatment as the control group. In addition, 1 L Hoagland nutrient solution was added with 1 mM KCl to balance K^+^ for phosphate deficiency treatment.

The Chinese fir plants with good growth after seedling survival were selected for phosphate deficiency and hormone treatment. Phosphate starvation treatment was divided into the normal group and the no phosphate group. These Chinese fir plants were treated for 0, 2, 4, 8, 12, and 24 h and 3, 5, and 7 days, respectively. The concentration of exogenous hormones was set as 0.1 mM ABA (abscisic acid) and 0.1 mM JA (jasmonic acid) [33], and short-term hormone treatment was performed on the roots of Chinese fir, and the treatment time was 0, 1, 3, 6, and 12 h. Three parts (root, stem, and leaf) of normal cultivated Chinese fir seedlings were used as materials, with the root as the control group. Five biological replicates were set for the above-mentioned group of experiments. After treatment, the root was collected and put in liquid nitrogen, then stored in the refrigerator at −80 °C.

Total RNA was extracted from the above samples and reverse-transcribed into cDNA. The expression pattern of *ClNAC100* was analyzed by RT-qPCR according to the PerfectStart Green qPCR SuperMix(+Dye II) kit (Nanjing, China) instructions, with the *ClActin* gene as an internal reference. Primers are listed in Appendix A, and the expression of the derived data was calculated using the 2^−△△CT^ method. Expression data from different treatments or parts were compared using Duncan’s method for multiple comparisons.

### 4.6. Promoter Cloning and Activity Analysis of ClNAC100

In total, 100 mg of well-grown Chinese fir root was selected as material, and genomic DNA was extracted according to the product description by TIANGEN’s Plant Genomic DNA Kit (Beijing, China). The purity of isolated genomic DNA was analyzed by running the samples in 0.5% agarose gel. ClNAC100 promoters were cloned from the genomic DNA of Chinese fir using TaKaRa’s Genome Walking Kit (Beijing, China). After gel electrophoresis verification and gel purification, the pMD-18 vector was connected to transform DH5α, and positive clones were selected for propagation and sent to the sequencing company for sequencing.

The software TSSP 5.1 and lab tools were used to predict the promoter transcription start site (TSS) of the ClNAC100, and the range of promoter core region was narrowed according to the highest score. Then, PlantCARE was used to analyze the cis-acting elements in the promoter sequence.

The isolated *ClNAC100* promoter and its truncations were cloned into the plant expression vector pBI121 using the primer-specific restriction sites 5′-HindIII and BamHI-3′, which replace the 35 s promoter. The vector pBI121 and recombinant vectors were then transformed into *Agrobacterium tumefaciens* strain GV3101 and confirmed using colony PCR. The positive agrobacterium was propagated, 3–4 week-old tobacco seedlings with good growth were selected, and the bacterial solution was injected into the lower epidermis of tobacco leaves using a 1 mL syringe. At the end of infection, tobacco leaves were stained with GUS, decolorized with 75% alcohol, observed, and photographed.

### 4.7. Yeast One-Hybrid

The promoter fragment of *ClNAC100* was divided into 400~700 bp fragments according to the distribution of cis-elements and inserted into the upstream of the AbAir reporter gene of pbaite-AbAi vector to construct pro1-pAbAi, pro2-pAbAi, and pro3-pAbAi recombinant vectors. BbsB I-linearized recombinant bait plasmids were integrated into the genome of Y1HGold. The transformed competent cells were then transferred onto solid agar SD/-Ura and incubated for 3 days. Single clones were then identified via colony PCR using Matchmaker Insert CheckPCR Mix 1 (Clontech Shanghai, China).

Healthy yeast colonies grown in SD medium-Ura were selected and resuspended in 0.9% NaCl. After adjusting to obtain OD_600nm_ values of approximately 0.002, the bacterial solutions were plated on SD/-Ura plates containing different concentrations of AbA. The colonies were allowed to grow for 2–3 days at 30 °C. The minimum AbA concentration that completely inhibited colony growth was determined and used for further library screening.

The fresh competent cells prepared from the positive strains transformed with 5 μg of Chinese fir cDNA yeast library plasmid (previously stored in our laboratory) were painted with SD/-Leu/AbA plate at the corresponding concentration, and each plate was painted with 100 μL bacterial solution, then incubated at 30 °C for 3–5 days. Positive clones were relined on SD/-Leu/AbA medium at the corresponding concentration to generate new monoclonals. Clones that could grow normally were selected, and the inserted cDNA fragments in positive clones were amplified according to the instructions of the Matchmaker Insert Check PCR Mix 2 kit (TaKaRa). A total of 5 μL of the PCR product was subjected to electrophoretic analysis on a 1% agarose gel and the products with a band less than 750 bp were screened out, and the other products were sent to sequencing companies for sequencing. The duplicated genes were removed from the sequencing results, and the proteins were annotated by the blast function of NCBI and TAIR, then the hypothetical and uncharacterized proteins were removed.

### 4.8. Yeast Two-Hybrid

Yeast two-hybrid screening library using the Mating method with the N-terminal NAC domain which did not have transcriptional activation. The colonies that grew well on SD/-Trp/-Leu plates were diluted with 0.9%NaCl solution and plated on SD/-Trp/-Leu/-His/-Ade/X-α-gal plates. The negative (pGBKT7-Lam+ pGADT7-T) and positive controls (pGBKT7-53+ pGADT7-T) were used as references. The blue clones were selected for PCR amplification and verification. The products with a band size of less than 750 bp were wiped off, and the other products were sent to sequencing companies for sequencing. The sequencing results were compared with the blast function of NCBI and TAIR.

## 5. Conclusions

In conclusion, *ClNAC100* is a transcription factor that can be induced by Pi deficiency. It is positively regulated by ABA and negatively regulated by JA and is highly expressed in the roots and leaves of Chinese fir. The ClNAC100 promoter contains a large number of stress-, hormone-, and growth-related cis-elements, and its core promoter is located between −1519 bp and −589 bp. *ClNAC100* is regulated by AHL and TIFY family transcription factors as well as calmodulin, and its protein can interact with E3 ubiquitin ligase. These results will help us to further investigate the regulatory mechanism of *ClNAC100* under Pi starvation in the future. In particular, we screened some important factors that may be involved in the upstream regulatory pathways of *ClNAC100* in response to Pi deficiency stress. These results will point out the direction for our future research on the regulatory pathways of *ClNAC100* in response to Pi deficiency stress.

## Figures and Tables

**Figure 1 ijms-24-10486-f001:**
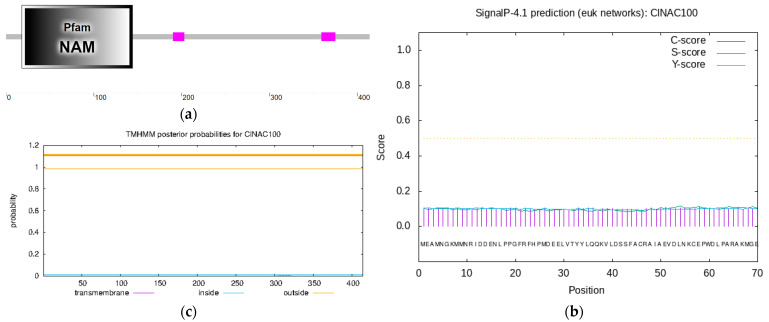
Bioinformatics analysis of the gene *ClNAC100*: (**a**) Prediction of ClNAC100 coding protein domain. (**b**) Prediction of ClNAC100 protein signal peptide. (**c**) Prediction of ClNAC100 protein transmembrane domain. (**d**) Sequence alignment of ClNAC100 and NACs from other plants, and the black box represents 100% homology. * means the intermediate value scale between two numerical scales.

**Figure 2 ijms-24-10486-f002:**
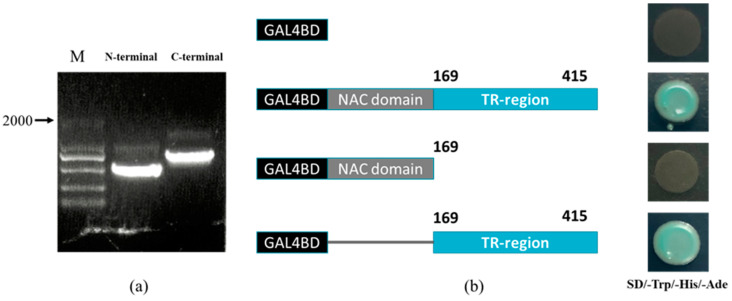
Analysis of *ClNAC100* transcriptional activity: (**a**) Cloning of *ClNAC100* NAC domain and TR-region. (**b**) Validation of transcriptional autoactivation of *ClNAC100*: different truncations of *ClNAC100* fused to the GAL4 DNA binding domain in pGBKT7 vector. Transactivation activity assay of *ClNAC100* in yeast strain Y2H Gold on the SD/-Trp/-Leu/-Ade/X-α-gal medium.

**Figure 3 ijms-24-10486-f003:**
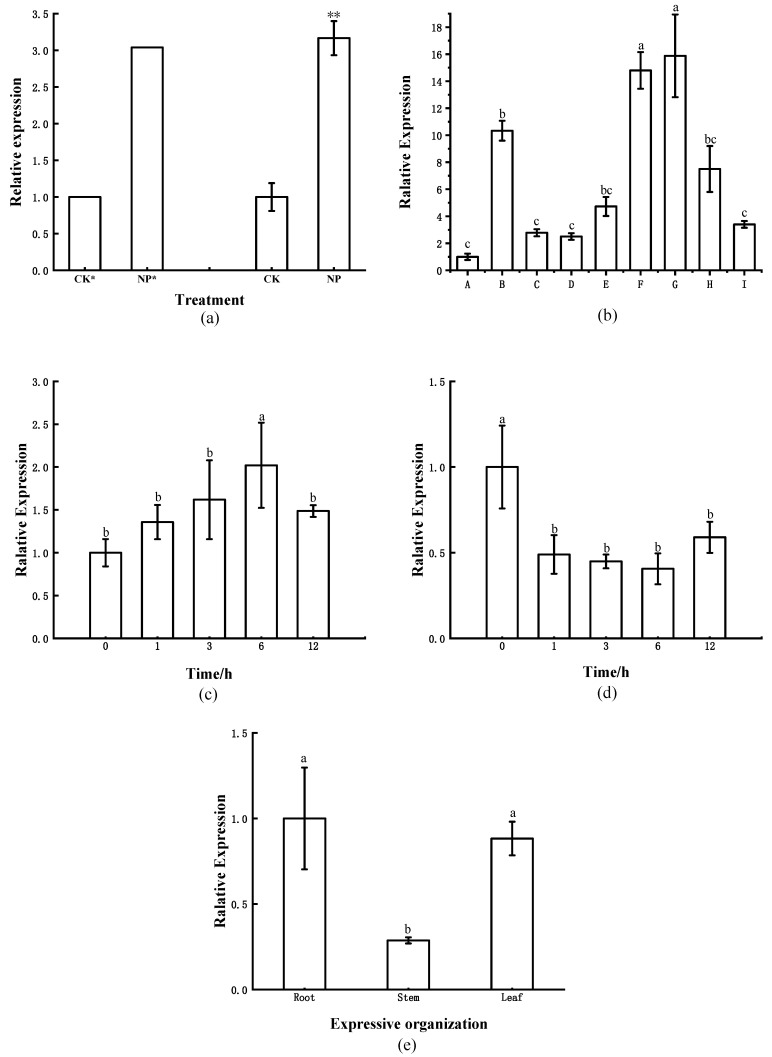
Expression profile of *ClNAC100*: (**a**) Expression of *ClNAC100* in the Pi-deficient treatment transcriptome of Chinese fir root (CK*–NP*), *ClNAC100* expression in the roots of Chinese fir under the same treatment as the Pi-deficient transcriptome (CK-NP, ** *p* < 0.01). (**b**) Expression patterns of *ClNAC100* at different phosphate deficiency times (A~I respectively represent 0 h (CK), 2 h, 4 h, 8 h, 12 h, 1 d, 3 d, 5 d, 7 d. (**c**) Expression patterns of ClNAC100 under ABA. (**d**) Expression patterns of ClNAC100 under JA. (**e**) Analysis of specific expression of *ClNAC100* in different parts of Chinese fir. Note: lowercase letters on the error line (a, b, and c) respectively represent significant differences between groups.

**Figure 4 ijms-24-10486-f004:**
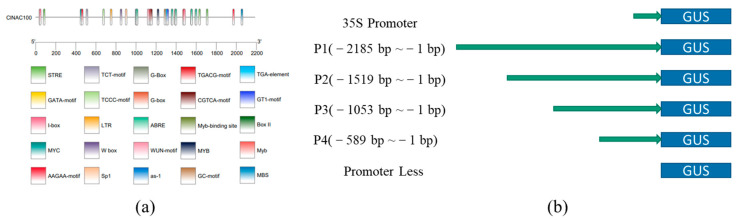
*ClNAC100* promoter analyses: (**a**) Visualization of *ClNAC100* promoter cis-element. (**b**) The expression vectors P1-PBI121, P2-PBI121, P3-PBI121, and P4-PBI121 with deletion of 5 promoters were constructed. (**c**) Analysis of promoter activity of *ClNAC100*: 35S-PBI121 is the positive control, no treatment (promoter less) is the negative control. The blue spots show the expression of the GUS gene driven by promoter fragments of different lengths in the vector on tobacco leaves.

**Figure 5 ijms-24-10486-f005:**
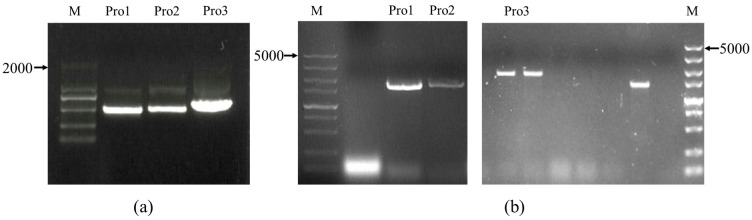
Insert promoter clone and validation of bait strains: (**a**) The cloned long promoter fragments were divided into three short fragments Pro1, Pro2, and Pro3 according to the cis-element distribution. (**b**) Identification of positive strains (amplified band size 1346 bp+ insert length), Pro1: pAbAi-Pro1 linearized plasmid transformed Y1H positive monoclonal PCR product; Pro2: pAbAi-Pro2 linearized plasmid transformed Y1H positive monoclonal PCR product; Pro3: pAbAi-Pro3 linearized plasmid transformed Y1H positive monoclonal PCR product.

**Figure 6 ijms-24-10486-f006:**
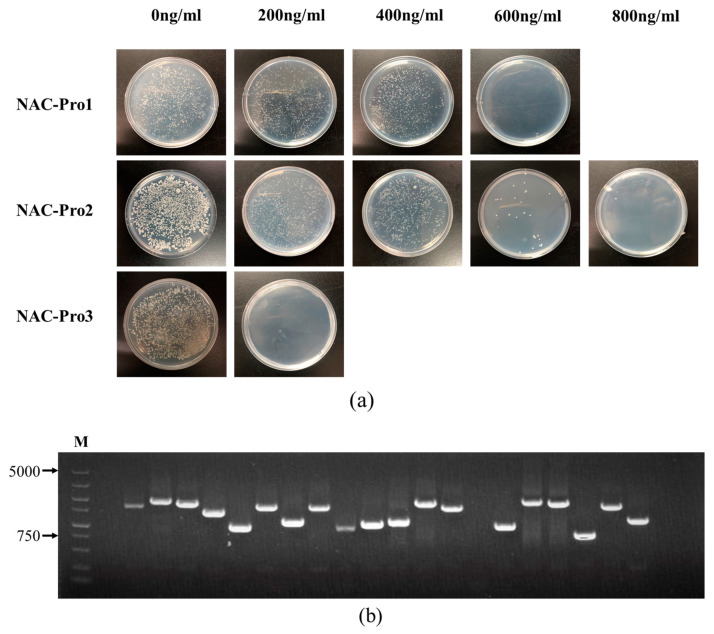
(**a**) Expression of AbAr gene in bait strain, identification of the minimum inhibitory concentration of AbA in different bait strains. (**b**) PCR validation of several colonies: bands of different sizes indicate different proteins to which the ClNAC100 promoter may bind.

**Table 1 ijms-24-10486-t001:** TSS position prediction.

Prediction Software	TSS Site	Score
TSSP	−1882 bp	0.6
labtool	−800 bp	1.208

**Table 2 ijms-24-10486-t002:** Annotation results of yeast one-hybrid screening protein.

Protein ID	Protein Annotation
Protein-No.1	N-carbamoyl-beta-alanine amidohydrolase
Protein-No.2	Cytochrome b561 and DOMON domain-containing protein At5g35735
Protein-No.3	Eukaryotic translation initiation factor 5A
Protein-No.4	Proton-coupled transporter that transports a wide
Protein-No.5	Prolyl aminopeptidase
Protein-No.6	Calcium-binding protein CML24
Protein-No.7	Glycerol-3-phosphate acyltransferase 9
Protein-No.8	Proteasome subunit alpha type-6
Protein-No.9	Caffeic acid 3-O-methyltransferase 1
Protein-No.10	(R, S)-reticuline 7-O-methyltransferase
Protein-No.11	PPIases accelerate the folding of proteins.
Protein-No.12	MLO-like protein 6
Protein-No.13	Cytochrome b6-f complex iron-sulfur subunit
Protein-No.14	AT-hook motif nuclear-localized protein10
Protein-No.15	Uracil-DNA glycosylase, mitochondrial
Protein-No.16	TIFY DOMAIN PROTEIN 8, TIFY8
Protein-No.17	Belongs to the cytochrome P450 family
Protein-No.18	Phenylalanine--tRNA ligase beta subunit

**Table 3 ijms-24-10486-t003:** Screening results of yeast two-hybrid and annotation results of proteins.

Protein ID	Protein Annotation
Protein-No.1	Heavy metal-associated isoprenylated plant protein 39
Protein-No.2	Protein translation factor SUI1 homolog
Protein-No.3	E3 ubiquitin-protein ligase RING
Protein-No.4	ATCBR, CBR, CBR1, NADH:CYTOCHROME B5 REDUCTASE 1
Protein-No.5	Potassium transporter 10

## Data Availability

Transcriptome data of phosphate deficiency in Chinese fir root (Accession number: PRJNA964476).

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
