# Peer review of "ClNAC100 Is a NAC Transcription Factor of Chinese Fir in Response to Phosphate Starvation"

_ijms, 2023, doi:10.3390/ijms241310486_

Round 1

Reviewer 1 Report

Please go through the attached file to get more specific comments. 

Overall it is a good research, with results leading toward conclusion. But the picture captions are not sufficient. Each caption should tell its own whole story. 

Few more lines should be added about the significance of the result. 

Sufficient experiments are done to publish it as a paper. But the researchers could have measured endogenous ABA and JA level during pi deficiency to support their claim that ClNAC100 is induced by ABA. 

Author Response

Thank you for taking time out of your busy schedule to review our manuscript“ClNAC100 is a NAC Transcription Factor of Chinese Fir in Response to Phosphate Starvation”(Manuscript ID: ijms-2441780). Now we have carefully corrected and replied to the manuscript for this revision. The suggestions from each reviewer were extremely helpful which led to the further improvement of our manuscript.

Please contact me if any questions. Thank you very much for your help.

We are looking forward to hearing from you soon.

Yours sincerely,

Peng Shuai, Ph. D

Chinese Fir Engineering Technology Research Center of the State Forestry and Grassland Administration

Fujian Agriculture and Forest University

Cangshan District, Fuzhou 350002, China

Point 1: Overall it is a good research, with results leading toward conclusion. But the picture captions are not sufficient. Each caption should tell its own whole story.

Response 1: As the Reviewer suggested that we have added more detailed descriptions to the pictures so that readers can better understand the content of the picture narrative. For this reason, more descriptions have been added to the titles of Figures 2(b)(P5/L167-169): Different truncations of ClNAC100 fused to the GAL4 DNA binding domain in pGBKT7 vector; Transactivation activity assay of ClNAC100 in yeast strain Y2H Gold on the SD/-Trp/-Leu/-Ade/X-α-gal medium. Figure 3(e)(P6/L203-204): Analysis of specific expression of ClNAC100 in different parts of Chinese fir. Figure 4(c)(P8/L233-235): 35S-PBI121 is the positive control, no treatment (promoter less) is negative control. The blue spots show the expression of the GUS gene driven by promoter fragments of different lengths in the vector on tobacco leaves. Figure 5 (a)(P8/L236-238): The cloned long promoter fragments were divided into three short fragments Pro1, Pro2 and Pro3 according to the cis-element distribution. Figure 5 (b)(P8/L238-241): Pro1: pAbAi-Pro1 linearized plasmid transformed Y1H positive monoclonal PCR product; Pro2: pAbAi-Pro2 linearized plasmid transformed Y1H positive monoclonal PCR product; Pro3: pAbAi-Pro3 linearized plasmid transformed Y1H positive monoclonal PCR product. Figure 6(a)(P9/L243-244): identification of the minimum inhibitory concentration of AbA in different bait strains. Figure 6(b)(P9/L244-245): bands of different sizes indicate different proteins to which the ClNAC100 promoter may bind.

Point 2: Few more lines should be added about the significance of the result.

Response 2: In response to the reviewer's suggestion, we have added a description of the significance of the results in the conclusion section at the end of the article. We've added content(P14/L464): “In particular, we screened some important factors that may be involved in the upstream regulatory pathways of ClNAC100 in response to Pi deficiency stress. These results will point out the direction for our future research on the regulatory pathways of ClNAC100 in response to Pi deficiency stress.”

Point 3: Sufficient experiments are done to publish it as a paper. But the researchers could have measured endogenous ABA and JA level during pi deficiency to support their claim that ClNAC100 is induced by ABA.

Response 3: This is a good suggestion and thank you for your suggestion. Following traditional validation of induction, we did not think of the measurement of endogenous ABA and JA levels. Moreover, due to the problems of time, space, and materials, we may not supplement this part of the data. We are willing to add experiments in this part in the future.

Reviewer 2 Report

In this manuscript, the authors identify and characterize a NAC transcription factor with increased expression under Pi deficiency. The results appear scientifically sound and supported by the presented evidence. However, the current state of presentation of the work is inadequate and requires more detail.

Generally, the captions for the figures do not adequately describe what is being shown in the figure.

Text in Figure 1 is too small to be legible.

The caption for Figure 4 is overlaid on top of the image for figure 5.

Figure 5(b) shows "PCR validation of several colonies," but nowhere is it indicated what size bands are to be expected or what the different sized bands mean.

The text refers to supplementary materials and tables which are not provided along with the manuscript.

Finally, there are a number of typos and language errors that require significant editing and polish.

There are a number of typos and language errors that require significant editing and polish. E.g., in the abstract: "Based on Chinses fir root tissue's Pi deficient transcriptome data..."

Author Response

Thank you for taking time out of your busy schedule to review our manuscript“ClNAC100 is a NAC Transcription Factor of Chinese Fir in Response to Phosphate Starvation”(Manuscript ID: ijms-2441780). Now we have carefully corrected and replied to the manuscript for this revision. The suggestions from each reviewer were extremely helpful which led to the further improvement of our manuscript.

Please contact me if any questions. Thank you very much for your help.

We are looking forward to hearing from you soon.

Yours sincerely,

Peng Shuai, Ph. D

Chinese Fir Engineering Technology Research Center of the State Forestry and Grassland Administration

Fujian Agriculture and Forest University

Cangshan District, Fuzhou 350002, China

Point 1: In this manuscript, the authors identify and characterize a NAC transcription factor with increased expression under Pi deficiency. The results appear scientifically sound and supported by the presented evidence. However, the current state of presentation of the work is inadequate and requires more detail.

Response 1: According to the Reviewer's suggestion, we have reviewed the manuscript again, and have supplemented and improved the details as much as possible. We supplemented the picture title description, which can be seen in Point 2. We have also revised the table title: Table 2(P7/L229): Annotation results of yeast one-hybrid screening protein. Table 3(P9/L246): Screening results of yeast two-hybrid and annotation results of proteins.

Point 2: Generally, the captions for the figures do not adequately describe what is being shown in the figure.

Response 2: We have made corrections according to the Reviewer's comments. We described and revised the figures’ titles in more detail so that they showed as much as possible of what the figures showed. Figures 2(b)(P5/L167-169): Different truncations of ClNAC100 fused to the GAL4 DNA binding domain in pGBKT7 vector; Transactivation activity assay of ClNAC100 in yeast strain Y2H Gold on the SD/-Trp/-Leu/-Ade/X-α-gal medium. Figure 3(e)(P6/L203-204): Analysis of specific expression of ClNAC100 in different parts of Chinese fir. Figure 4(c)(P8/L233-235): 35S-PBI121 is the positive control, no treatment (promoter less) is negative control. The blue spots show the expression of the GUS gene driven by promoter fragments of different lengths in the vector on tobacco leaves. Figure 5 (a)(P8/L236-238): The cloned long promoter fragments were divided into three short fragments Pro1, Pro2 and Pro3 according to the cis-element distribution. Figure 5 (b)(P8/L238-241): Pro1: pAbAi-Pro1 linearized plasmid transformed Y1H positive monoclonal PCR product; Pro2: pAbAi-Pro2 linearized plasmid transformed Y1H positive monoclonal PCR product; Pro3: pAbAi-Pro3 linearized plasmid transformed Y1H positive monoclonal PCR product. Figure 6(a)(P9/L243-244): identification of the minimum inhibitory concentration of AbA in different bait strains.

Point 3: Text in Figure 1 is too small to be legible.

Response 3: We have replaced the original picture with one with clearer pixels.

Point 4: The caption for Figure 4 is overlaid on top of the image for figure 5.

Response 4: We have adjusted Figure 5 position as suggested.

Point 5: Figure 5(b) shows "PCR validation of several colonies," but nowhere is it indicated what size bands are to be expected or what the different sized bands mean.

Response 5: We have added an illustration of the meaning of the different bands in Figure 6(b) title(P9/L244-255): PCR validation of several colonies: bands of different sizes indicate different proteins to which the ClNAC100 promoter may bind.

Point 6: The text refers to supplementary materials and tables which are not provided along with the manuscript.

Response 6: The supplementary materials and forms provided in the original manuscript were also submitted at the time of the initial submission of the manuscript. We contacted the editor to confirm whether the supplementary materials were submitted. If not, we will supplement the materials and forms.

Point 7: Finally, there are a number of typos and language errors that require significant editing and polish.

Response 7: Some language problems in the manuscript were revised as suggested. We corrected some problems in the abstract as suggested by the reviewer: 1. Based on Pi deficient transcriptome data of Chinses fir root(P1/L14). 2. Correction of tense: It was positively regulated by ABA and negatively regulated by JA and ClNAC100 was highly expressed in the roots and leaves of Chinese fir(P1/L17-19). 3. we reported that a significantly differentially expressed NAC transcription factor named ClNAC100, which was cloned from the Phosphate-deficient transcripts of Chinese fir root(P2/L87-89).

Reviewer 3 Report

Manuscript ID ijms-2441780 review

CINAC100 is a NAC Transcription Factor of Chinese Fir in Re­sponse to Phosphate Starvation

Yuxuan Zhao, Shuotian Huang, Lihui Wei, Meng Li, Tingting Cai, Xiangqing Ma and Peng Shuai

Phosphorus belongs to the group of macronutrients. In this regard, phosphorus deficiency disrupts phosphorus- and energy-dependent reactions in the plant. Deciphering the mechanisms of plant protection allows us to find ways to regulate the state of plants under stressful conditions. In this regard, the presented manuscript is relevant.

A convincing experimental basis was provided in the manuscript ijms-2441780.

The authors cited about 40% of recent publications (in the last 5 years).

The authors identified the ClNAC100 gene, which is a transcription factor in Cunninghamia lanceolata plants. A multilevel regulation of ClNAC100 gene expression was shown. This gene is more strongly expressed in roots and leaf than in stem under Pi deficiency. The gene expression was found to be positively regulated by ABA and negatively by jasmonic acid.

A comparative analysis of the amino acid sequence of proteins of 8 plant species was performed. Presence of a conserved domain at the N-terminus and homology of sequences at the C-terminus allowed the authors to assume similar functions of the proteins.

A domain at the C-terminus of the protein was found to serve as a transcriptional activator with transcriptional activation activity. A large number of cis-elements related to hormonal responses were detected, including a salicylic acid response element, an ethylene response element, and an auxin-related element. Light-sensitive elements, stress-related elements, and stress-related transcription factor binding sites were present on the promoter.

ClNAC100 is regulated by transcription factors of the AHL and TIFY families as well as by calmodulin, and its protein can interact with E3 ubiquitin ligase.

The study extends insights into the regulatory mechanism of CNAC100 under Pi starvation. Possible pathways of phosphorus deficiency signaling are analyzed.

Remarks on keywords:

In the Keywords "Chinese fir" should be replaced by the Latin species name Cunninghamia lanceolata. Р1/L28.

Remarks on the methodology:

1.       On the basis of what data the investigated hormone concentrations were chosen should be specified. Р12/L375

2.       It is not correct to call tissues (root, leaf and stem). For example, the root consists of many tissues (epidermis, cortex, endoderm, pericycle, phloem and xylem). There are 3 growth zones in the cortex: the division zone, the stretching zone, and the differentiation (suction) zone. Which part of the root was taken for the study? Gene expression of which region of the root is dominant and gives a signal of change in root morphogenesis? Р12/L377

3.       Explain why the root was chosen as the control group. Р12/L378

Remarks on the formatting of the references:

The list of references should be designed according to the requirements of the journal:

References 1, 2, 7, 11, 20, 22, 40, 51

Author Response

Thank you for taking time out of your busy schedule to review our manuscript“ClNAC100 is a NAC Transcription Factor of Chinese Fir in Response to Phosphate Starvation”(Manuscript ID: ijms-2441780). Now we have carefully corrected and replied to the manuscript for this revision. The suggestions from each reviewer were extremely helpful which led to the further improvement of our manuscript.

Please contact me if any questions. Thank you very much for your help.

We are looking forward to hearing from you soon.

Yours sincerely,

Peng Shuai, Ph. D

Chinese Fir Engineering Technology Research Center of the State Forestry and Grassland Administration

Fujian Agriculture and Forest University

Cangshan District, Fuzhou 350002, China

Point 1: In the Keywords "Chinese fir" should be replaced by the Latin species name Cunninghamia lanceolata. Р1/L28.

Response 1: We have changed the "Chinese fir" to “Cunninghamia lanceolata” as suggested by the reviewer(P1/L28).

Point 2: On the basis of what data the investigated hormone concentrations were chosen should be specified. Р12/L375

Response 2: The selection of hormone concentration was based on our lab's previous studies on exogenous hormone treatment of other transcription factors of Chinese fir and some relevant references. We added reference 33 in the Methods section of the manuscript as the reference for the selection of hormone concentration(P12/L387).

Point 3: It is not correct to call tissues (root, leaf and stem). For example, the root consists of many tissues (epidermis, cortex, endoderm, pericycle, phloem and xylem). There are 3 growth zones in the cortex: the division zone, the stretching zone, and the differentiation (suction) zone. Which part of the root was taken for the study? Gene expression of which region of the root is dominant and gives a signal of change in root morphogenesis? Р12/L377

Response 3: This is such an important suggestion that we made an amendment to the manuscript. Chinese fir seedlings were selected for the experiment. The roots of Chinese fir seedlings are thin and tender, and we use complete roots of Chinese fir seedlings for RNA extraction. Accordingly, We changed the “root tissue” to “root”(P6/L199, P12/L389, P12/L392).

Point 4: Explain why the root was chosen as the control group. Р12/L378

Response 4: Root was chosen as the control group because the root is an important organ for sensing Pi deficiency of Chinese fir and ClNAC100 is expressed in Chinese fir roots. Therefore, we used the relative expression of ClNAC100 in roots as a control group to compare the relative expression of ClNAC100 in other parts of Chinese fir.

Point 5: The list of references should be designed according to the requirements of the journal: References 1, 2, 7, 11, 20, 22, 40, 51

Response 5: We checked the suggested references and other references and revised the problematic parts(P14/L485, P14/488, P14/497, P14/507, P15/L528, P15/L531, P15/L569, P16/L592).
